# Effects of Mannanoligosaccharide Supplementation on the Growth Performance, Immunity, and Oxidative Status of Partridge Shank Chickens

**DOI:** 10.3390/ani9100817

**Published:** 2019-10-16

**Authors:** Minyu Zhou, Yuheng Tao, Chenhuan Lai, Caoxing Huang, Yanmin Zhou, Qiang Yong

**Affiliations:** 1Jiangsu Co-Innovation Center of Efficient Processing and Utilization of Forest Resources, College of Chemical Engineering, Nanjing Forestry University, Nanjing 210037, China; belle1234@163.com (M.Z.); tyh0305@njfu.edu.cn (Y.T.); lch2014@njfu.edu.cn (C.L.); hcx@njfu.edu.cn (C.H.); 2Key Laboratory of Forestry Genetics & Biotechnology (Nanjing Forestry University), Ministry of Education, Nanjing 210037, China; 3College of Animal Science and Technology, Nanjing Agricultural University, Nanjing 210095, China; zhouym6308@163.com

**Keywords:** mannanoligosaccharide, growth performance, immunity, oxidative status, Partridge Shank chickens

## Abstract

**Simple Summary:**

To keep animals healthy and maintain sustainability, modern poultry production industry uses functional feed additives such as mannanoligosaccharides to minimize the potential threat of disease and protect the intestinal mucosa against invading microorganisms. However, most of them are obtained by chemical synthesis that may cause environmental pollution. Thus, we found a way to produce mannanooligosaccharides by an enzyme called β-mannanase to avoid pollution. This enzyme is produced by the fungus species *Aspergillus niger*. In the present study, we evaluated such enzymatic mannanooligosaccharide and found it can improve oxidative status and immunity in broiler chickens.

**Abstract:**

Mannanoligosaccharides (MOS) can be used in poultry production to modulate immunity and improve growth performance. So, we hypothesized that our enzymatic MOS could achieve the same effects in broilers. To investigate this, a total of 192 one-day-old Partridge Shank chickens were allocated to four dietary treatments consisting of six replicates with eight chicks per replicate, and they were fed a basal diet supplemented with 0, 0.5, 1 and 1.5 g MOS per kg of diet(g/kg) for42 days. Treatments did not affect the growth performance of chickens. Dietary MOS linearly increased the relative weight of the bursa of Fabricius and jejunal immunoglobulin M (IgM) and immunoglobulin G (IgG) content, whereas it linearly decreased cecal *Salmonella* colonies at 21 days (*p* < 0.05). The concentration of jejunal secretory immunoglobulin A (sIgA) and IgG at 42 days as well as ileal sIgA, IgG, and IgM at 21 and 42 days were quadratically enhanced by MOS supplementation (*p* < 0.05). Also, chickens fed MOS exhibited linear and quadratic reduction in jejunal malondialdehyde (MDA) accumulation (*p* < 0.05). In conclusion, this enzymatic MOS can improve the immune function and intestinal oxidative status of Partridge Shank chickens.

## 1. Introduction

Oligosaccharides, such as mannanoligosaccharides (MOS) are now widely used as functional feed additives in modern poultry production. MOS are indigestible to monogastric animals and can inhibit colonization of pathogenic microorganisms in the intestinal tract by binding pathogenic bacteria that possess mannose-specific type-I fimbriae and by its prebiotic activity. At the other hand, MOS have been found to enhance the growth of some probiotics such as cecal *Lactobacillus* species and *Bifidobacterium* species.

Extensive reports have proved that dietary MOS supplementation can enhance immunity and intestinal health, resulting in better growth performance of animals under both normal and adverse conditions [1,2,3,4,5,6]. Additionally, some exciting findings on MOS research have currently been observed by Bozkurt et al. [7], Attia et al. [8] and Zheng et al. [9], who have shown that dietary MOS addition can act as a free radical scavenger to improve the body’s antioxidant capacity through inhibiting lipid peroxidation and/or elevating antioxidant enzymes activities in laying hens, broilers, and sheep. Furthermore, Liu et al. [10] have reported that the inclusion of dietary MOS can relieve hepatic oxidative damage of fish under adverse conditions. It has been demonstrated that dietary MOS supplementation increases water-holding capacity and tenderness [8,11], whereas it decreases the fat content of muscle in animals [8,12]. In a published paper, Zhang et al. [13] illustrated that dietary yeast cell wall inclusion, a widely used MOS product, reduced the concentration of malondialdehyde (MDA), an end-product of lipid peroxidation, in raw and boiled muscles in broilers.

MOS originates from different sources, and it has been repeatedly reported that various mannanases from bacteria, fungi, and plants can hydrolyze different mannan-containing polysaccharides to yield MOS [14,15,16,17,18,19,20,21,22]; however, the supply of MOS is not adequate to meet the demand. So, an economically viable technique for producing MOS has yet to be identified and developed. *Amorphophallus konjac* K. Koch is an underutilized agricultural material with low commercial value in China where it is typically used as animal feed and as a gelling and thickening ingredient for human foods [23]. It has been recognized as a safe material according to the FDA (Food and Drug Administration) [24]. Almost 60% of konjac is glucomannan, a previously noted precursor to MOS. The glucomannan from *Amorphophallus konjac* (KGM) and MOS from glucomannan consist of a linear chain of β-1,4-d-glucose and d-mannose. Structural studies of MOS from KGM revealed that it contains only glucose and mannose at a molar ratio of 1:1.6 [23]. In addition, it was found that branching occurs at β-1,6- glucoses approximately three times for every 32 sugar residues [25]. Finally, it has been found that most MOS has a degree of polymerization (DP) between 2 and 6. Little is known about the effect of this MOS on broilers, especially Partridge Shank chickens, an important local chicken breed. We hypothesized that the MOS would exhibit a high bioavailability in vivo. The current study was therefore conducted to evaluate the effects of enzymatic MOS from KGM on the growth performance, immunity, and antioxidant status of Partridge Shank chickens.

## 2. Materials and Methods

The experimental procedures used in this study were approved by the Nanjing Agricultural University Institutional Animal Care and Use Committee. The ethical code is NJAU20171104.

### 2.1. Mannanoligosaccharide

Mannanoligosaccharide (MOS) was prepared from KGM produced by the laboratory using enzymatic hydrolysis. The KGM used in this experiment was prepared from *Amorphophallus konjac* bought from the local market of Yunnan Province of China. The enzyme used was β-mannanase produced from *Aspergillus niger* by the laboratory. Hydrolysis was performed for 2 h at pH 5.0 with an environmental temperature of 50 °C. Post hydrolysis, enzymatic hydrolysate was free flowing. The enzyme activity was inactivated by putting enzymatic hydrolysate in a beaker into boiling water for 10 min, then ultrafiltration was used to separate the impurities to get MOS. Finally, spray drying (BUCHI, Flawil, Switzerland) was used to prepare solid MOS.

### 2.2. Husbandry, Diets and Experimental Design

A total of one hundred and ninety-two one-day-old Partridge Shank chicks with similar initial weight obtained from a commercial hatchery were randomly allocated into four dietary treatments. Each treatment included 48 chicks that consisted of six replicates (one cage per replicate). Birds in the four treatments were fed a basal diet supplemented with 0, 0.5, 1 and 1.5 g MOS per kg of diet for 42 days. Ingredient composition and nutrient content of the basal diets are presented in Table 1. Birds had free access to mash feed and water in three-level cages (120 cm × 60 cm × 50 cm; 0.09 m^2^ per chick) in a temperature-controlled room with continuous lighting. The temperature of the room was maintained at 32 to 34 °C for the first 3 days and then reduced by 2–3 °C per week to a final temperature of 26 °C. At 21 days and 42 days of age, birds were weighed after feed deprivation for 12 h and feed intake was recorded by replicate (cage) to calculate average daily feed intake (ADFI), and average daily gain (ADG). Birds that died during the experiment were weighed, and the data were included in the calculation of feed conversion ratio (FCR).

### 2.3. Sample Collection

At 21 and 42 days, one bird (close to the average body weight of birds in each cage) from each replicate (48 birds in total) was selected and weighed after feed deprivation for 12 h. After that, blood samples (around 5 mL each) were taken from the wing vein and centrifuged at 4450× *g*, 15 min at 4 °C to separate serum, which was frozen at −20 °C until analysis. After blood collection, the chickens were euthanized by cervical dislocation and immediately necropsied. Following necropsy, the whole gastrointestinal tracts were quickly removed. Bursa of Fabricius, thymus, and spleen were then collected and weighed to calculate the relative organ weights using the following formula: relative weight of immune organ (g/kg) = immune organ weight (g)/body weight (kg). Jejunum (from the end of the pancreatic loop to the Meckel’s diverticulum) and ileum (from Meckel’s diverticulum to the ileocecal junction) were then excised free of the mesentery and placed on a chilled stainless-steel tray. The jejunal, and ileal mucosa were scratched carefully using a sterile glass microscope slide, which were then rapidly frozen in liquid nitrogen and stored at −80 °C for further analysis. Then cecum samples were quickly removed aseptically, and cecal contents were cultured to determine the population of *Lactobacillus*, *Salmonella* and *Escherichia coli*.

### 2.4. Microflora Population Measurement

Approximately 0.2 g of aseptically removed cecal contents were diluted in 2 mL of sterilized saline (154 mmol/L), and then three 10-fold serial dilutions were made from the diluted cecal contents (10^−3^, 10^−4^ and 10^−5^ for *Salmonella*; 10^−4^, 10^−5^ and 10^−6^ for *Escherichia coli* and *Lactobacillus*). A 100 μL portion of the last three dilutions were then spread evenly onto plates. *Escherichia coli* colonies were enumerated on MacConkey agar (Qingdao Hope Bio-Technology Co. Ltd., Qingdao, Shandong, China) at 37 °C for 24 h. *Lactobacillus* were enumerated on MRS agar (Qingdao Hope Bio-Technology Co. Ltd., Qingdao, Shandong, China) medium at 37 °C for 48 h. *Salmonella* colonies were determined on Bismuth sulfite agar (Qingdao Hope Bio-Technology Co. Ltd., Qingdao, Shandong, China) and incubated at 37 °C for 24 h. All plates with countable colonies were enumerated and averaged to express log CFU (Colony-Forming Units) per gram of cecal content.

### 2.5. Determination of Mucosal Immune and Antioxidant Parameters

Approximately 0.3 g mucosal samples from jejunum and ileum were homogenized (1:9, wt/vol) with ice-cold 154 mmol/L sodium chloride solution using an Ultra-Turrax homogenizer (Tekmar Co., Cincinatti, OH, USA) and then centrifuged at 4450× *g* for 15 min at 4 °C. The supernatant was then collected and stored at −20 °C for subsequent analysis.

Total superoxide dismutase (T-SOD) activity, and malondialdehyde (MDA) content were analyzed using commercial diagnostic kits (Nanjing Jiancheng Bioengineering Institute, Nanjing, Jiangsu, China) according to the manufacturer’s instructions. The activity of T-SOD was analyzed by the hydroxylamine method [26], and one unit of T-SOD was defined as the amount of enzyme per milliliter of mucosa required to produce 50% inhibition of the rate of nitrite production at 37°C. MDA concentration was measured by barbiturate thiosulfate assay [27], and was expressed as nanomole per milliliter of mucosa.

Concentrations of immunoglobulin M (IgM), immunoglobulin G (IgG), and secretory immunoglobulin A (sIgA) were measured in appropriately diluted mucosal samples by enzyme-linked immunosorbent assay (ELISA) using microtiter plates and chicken-specific IgM, IgG, sIgA ELISA quantitation kits (Nanjing Jiancheng Bioengineering Institute, Nanjing, Jiangsu, China). All results were normalized against total protein concentration in each sample for inter-sample comparison. Finally, total protein concentration was determined by using a total protein quantitation kit (Nanjing Jiancheng Bioengineering Institute, Nanjing, Jiangsu, China).

### 2.6. Statistical Analysis

Data was analyzed by one-way analysis of variance (ANOVA) using SPSS statistical software (Ver. 19.0 for windows, SPSS Inc., Chicago, IL, USA). The replicate (cage) was defined as the experimental unit. Polynomial contrasts were used to test the linear and quadratic effects of MOS levels. The level of significance was *p* < 0.05 in all analyses. Results are presented as means alongside their pooled standard errors of means.

## 3. Results

### 3.1. Growth Performance

Chickens given basal diets supplemented (Table 2) with MOS exhibited similar growth performance compared with the control group during the 42-day study (*p* > 0.05).

### 3.2. Realtive Immune Organ Weights

As shown in Table 3, the inclusion of MOS quadratically increased the relative weight of bursa of Fabricius at 21 days (*p* < 0.05), but this effect was not observed at 42 days (*p* > 0.05). Also, the relative weights of the thymus and spleen were not altered by the MOS diet (*p* > 0.05).

### 3.3. Cecal Microflora Population

In Table 4, it can be seen that MOS had a linear effect on *Salmonella* colonies (*p* < 0.05) in the cecal content at 21 days. However, cecal *Escherichia coli* and *Lactobacillus* colonies were not affected by MOS supplementation during the whole experiment (*p* > 0.05).

### 3.4. Intestinal Immunoglobulins Contents

Chickens exhibited similar content of sIgA in the jejunal mucosa among groups at 21 days (Table 5, *p* > 0.05). MOS linearly increased jejunal IgM and IgG contents (*p* < 0.05) at 21 days and quadratically increased jejunal sIgA and IgG levels at 42 days *(p* < 0.05). Simultaneously, ileal sIgA, IgM and IgG contents were quadratically increased in 42 days (*p* < 0.05).

### 3.5. Intestinal Oxidative Status

As shown in Table 6, chickens fed MOS exhibited linear and quadratic reduction in jejunal MDA accumulation at 21 days (*p* < 0.05), and quadratic effect on ileal MDA content at 42 days (*p* < 0.05). However, intestinal SOD activity was similar among treatments (*p* > 0.05).

## 4. Discussion

### 4.1. Growth Performance

Sims et al. [28] and Attia et al. [29] demonstrated that dietary MOS supplementation can improve the growth performance of poultry under normal conditions. In broilers, Geier et al. [30] found that when broiler feed contained MOS, the growth performance of broilers was unchanged. This study demonstrated that MOS supplementation exerted no significant effect on the growth performance of broilers, and this was consistent with the findings of Munyaka et al. [31], who reported that dietary supplementation with yeast-derived MOS preparation did not alter growth performance and mortality in broilers. In contrast, Churchil et al. [32] observed that yeast-derived MOS inclusion increased the body weight of broilers. In addition, Gao et al. [33] demonstrated that the growth performance of broilers was optimized by adding the yeast-derived MOS. Therefore, the unchanged growth performance observed in this study may be associated with the source of MOS used as the dietary supplement; that is, the broilers may digest less nutrients from our MOS. Based on this result, further studies are needed to evaluate the influences of different sources of MOS on the growth performance of chickens, and to evaluate how to further process our MOS so that it can increase nutrient digestibility of chickens.

### 4.2. Relative Immune Organ Weights

Relative immune organ weights could partially reflect the development and growth of immune organs. The current study showed that MOS quadratically increased the relative weight of bursa of Fabricius at 21 days, which plays a vital role in development and maturation of B-lymphocytes and the diversification of specific antibodies [34]. Thus, MOS supplementation may increase the weight of bursa by stimulating the proliferation of bursal lymphocytes. Also, digestive microbial antigen stimulation plays a vital role in the development of lymphoid organ tissue [35]. Li et al. [36] reported that the increased weight of bursa may be associated with possible changes to the intestinal microorganism population induced by yeast derived MOS supplementation. Dietary MOS supplementation, therefore, represents a nutritional strategy that could favor intestinal colonization of beneficial bacteria, thereby conferring intestinal health benefits to the host. Further study is required to verify this conjecture.

### 4.3. Cecal Microflora Population

MOS in this experiment is a plant-derived oligosaccharide, which can promote the growth of *Bifdobacteria*, which decreases colonization by enteric pathobionts like *Salmonella* and *Escherichia coli*, regulates immune signaling, and improves mucosal integrity [37,38]. It is well documented that MOS competitively adsorbs to the mannosespecific type 1 fimbriae of *Escherichia coli* and other pathogens, thereby limiting their colonization of the intestinal epithelium. This phenomenon results in the pathogens ultimately being excreted from the intestine [39,40]. Muthusamy et al. [41] reported that dietary MOS lowered *Salmonella* spp. and *Escherichia coli* number in the small intestine (duodenum, jejunum and ileum) of broilers with poor health or *Salmonella* challenged. In this study, MOS had a linear decreasing effect on *Salmonella* colony in the cecal content at 21 days, indicating that the prepared MOS can decrease colonization by enteric pathobionts. Different results were found by Li et al. [36] whereby MOS supplementation did not alter *Escherichia coli* and *Salmonella* colonies in the cecal content (only a decreased tendency was noted). Thus, oligosaccharides from different sources and different chain lengths may have different results on different intestinal microorganisms. This hypothesis requires further research to prove it.

### 4.4. Intestinal Immunoglobulins

The immune system guards the body against foreign substances and protects it from invasion by pathogenic organisms. In chickens, three classes of immunoglobulins participate in immune system maintenance. These immunoglobulins have been identified as IgM, IgG and IgA [42]. sIgA plays an important role in the protection and homeostatic regulation of intestinal mucosal epithelia separating the outside environment from the inside of the body. The primary function of sIgA is referred to as immune exclusion, a process that limits the access of numerous microorganisms and mucosal antigens to the thin and vulnerable mucosal barriers [43]. Savage et al. [44] reported that when feeding MOS to broilers, the concentration of IgA in the bile increased 14.2%, and that the MOS may have a mechanism that directly protects the mucosa. The present study showed that MOS linearly increased jejunal IgM and IgG contents at 21 days, while it quadratically increased sIgA and IgG contents at 42 days. Simultaneously, ileal sIgA, IgM and IgG contents were quadratically increased at 42 days. Similar results were also observed by Li et al. [36] and Gao et al. [33]. We assumed that the main target of the prepared MOS is located in the intestine, and it may simulate the development of intestinal cells in the jejunum and ileum to secrete more immunoglobulins. This result indicates that the prepared MOS can improve intestinal immune status.

### 4.5. Intestinal Oxidative Status

Reactive oxygen species (ROS) are produced during normal metabolism in cells, but concentration of ROS exceeding the antioxidant protection levels of cells can cause widespread damage to DNA, proteins and endogenous lipids [45]. SOD is generally regarded as one of the main antioxidant enzymes in scavenging the oxygen free radical [46]. The MDA is the main end product of lipid peroxidation by ROS, and increased MDA accumulation is an important indication of lipid peroxidation [47]. MOS from konjac has been reported to display relatively good antioxidative properties [48]. In poultry, enhanced SOD activity in the serum of broilers fed dietary MOS has recently been found by Attia et al. [49]. Bozkurt et al. [7] reported that dietary MOS supplementation could decrease MDA concentration in both eggs and liver, and increase SOD activity in the liver in laying hens. In this study, MOS linearly and quadratically decreased jejunal MDA accumulation in 42 days and it had quadratic effect on ileal MDA accumulation at 42 days. This was in agreement with the results of Liu et al. [10], who demonstrated that dietary MOS inclusion decreased MDA accumulation in fish under adverse conditions. According to the literature, dietary MOS supplementation can accelerate gastrointestinal maturation and increase nutrient absorption for better growth performance in organisms [50,51,52], which may simultaneously and indirectly contribute to improving the adsorption and utilization of small molecules related to the synthesis of antioxidants. Thus, in the current study, elevated oxidative status in the intestinal mucosa by MOS supplementation might also be related to the promotion of MOS addition on the gut ecology and digestive function in animals [51,52].

## 5. Conclusions

In this study, MOS did not affect growth performance whereas it improved immune function (enhanced relative weight of bursa of Fabricius, enhanced jejunal sIgA and IgG contents and ileal sIgA and IgG levels), intestinal oxidative status (decreased jejunal MDA content), and regulated the cecal microflora population (reduced cecal *Salmonella* population) in Partridge Shank chickens.

## Figures and Tables

**Table 1 animals-09-00817-t001:** Composition and nutrient level of basal diet (g/kg, as-fed basis unless otherwise stated).

Items	1–21 Days	22–42 Days
Ingredients		
Corn	576.1	622.7
Soybean meal	310	230
Corn gluten meal	32.9	60
Soybean oil	31.1	40
Limestone	12	14
Dicalcium phosphate	20	16
L-Lysine·HCL	3.4	3.5
DL-Methionine	1.5	0.8
Sodium chlodire	3	3
Premix ^1^	10	10
Calculated nutrient levels ^2^		
Apparent metabolizable energy (MJ/kg)	12.56	13.19
Crude protein	211	196
Calcium	10.00	9.50
Available phosphorus	4.60	3.90
Lysine	12.00	10.50
Methionine	5.00	4.20
Methionine + cysteine	8.50	7.60
Analyzed composition ^3^		
Crude protein	208	192
Ash	57.2	56.5

^1^ Premix provided per kilogram of diet: vitamin A (transretinyl acetate), 10,000 IU; vitamin D3 (cholecalciferol), 3000 IU; vitamin E (all-rac-α-tocopherol), 30 IU; menadione, 1.3 mg; thiamin, 2.2 mg; riboflavin, 8 mg; nicotinamide, 40 mg; choline chloride, 600 mg; calcium pantothenate, 10 mg; pyridoxine·HCl, 4 mg; biotin, 0.04 mg; folic acid, 1 mg; vitamin B12 (cobalamin), 0.013 mg; Fe (from ferrous sulphate), 80 mg; Cu (from copper sulphate), 8.0 mg; Mn (from manganese sulphate), 110 mg; Zn (from zinc oxide), 60 mg; I (from calcium iodate), 1.1 mg; Se (from sodium selenite), 0.3 mg; ^2^ the nutrient levels were as fed basis; ^3^ Values based on analysis of triplicate samples of diets.

**Table 2 animals-09-00817-t002:** Growth performance of Partridge Shank chickens fed diets supplemented with or without mannanoligosaccharide (MOS).

Items	Control	0.5 g/kg MOS	1 g/kg MOS	1.5 g/kg MOS	SEM	*p*-Value	
						L	Q
ADG (g/days)							
1–21days	17.31	16.50	16.72	16.92	0.151	0.489	0.102
22–42days	43.81	43.43	43.63	43.37	0.340	0.732	0.939
1–42days	32.68	32.12	32.33	32.26	0.221	0.617	0.601
ADFI (g/days)							
1–21days	27.04	25.89	26.72	25.74	0.246	0.156	0.860
22–42days	101.42	93.65	102.35	98.52	0.944	0.999	0.158
1–42days	69.55	64.78	69.99	67.41	0.580	0.753	0.204
FCR (g:g)							
1–21days	1.56	1.57	1.60	1.52	0.011	0.246	0.052
22–42days	2.31	2.16	2.35	2.27	0.020	0.605	0.191
1–42days	2.13	2.02	2.17	2.09	0.015	0.646	0.438

MOS = mannanoligosaccharide; ADG = average daily gain; ADFI = average daily feed intake; FCR = feed conversion ratio; SEM = standard error of means (each treatment included 48 chickens and consisted of 6 replicates); L = linear; Q = quadratic.

**Table 3 animals-09-00817-t003:** Immune organ weights from Partridge Shank chickens fed diets supplemented with or without MOS (g/kg).

Items	Control	0.5 g/kg MOS	1 g/kg MOS	1.5 g/kg MOS	SEM	*p*-Value	
						L	Q
Thymus							
Days 21	1.10	1.07	1.06	1.10	0.06	0.976	0.785
Days 42	2.48	3.78	3.27	2.51	0.26	0.856	0.055
Spleen							
Days 21	0.83	0.74	0.73	0.91	0.04	0.546	0.132
Days 42	4.13	3.14	5.13	4.62	0.30	0.180	0.677
Bursa of Fabricius							
Days 21	0.98	1.69	1.34	1.42	0.07	0.110	0.031
Days 42	1.57	1.71	1.31	1.62	0.14	0.854	0.768

MOS = mannanoligosaccharide; relative immune organ weight that was expressed relative to body weight; SEM = standard error of means (each treatment included 48 chickens and consisted of 6 replicates); L = linear; Q = quadratic.

**Table 4 animals-09-00817-t004:** Microflora population in the cecal content of Partridge Shank chickens fed diets supplemented with or without MOS (log CFU/g content).

Items	Control	0.5 g/kg MOS	1 g/kg MOS	1.5 g/kg MOS	SEM	*p*-Value	
						L	Q
*Escherichia coli*							
Days 21	7.98	8.04	7.97	8.45	0.15	0.360	0.514
Days 42	6.89	6.94	6.68	6.04	0.20	0.946	0.820
*Salmonella*							
Days 21	8.41	7.73	7.10	7.44	0.18	0.028	0.126
Days 42	6.16	6.35	6.11	6.52	0.18	0.652	0.777
*Lactobacillus*							
Days 21	8.41	8.18	8.51	8.06	0.08	0.326	0.501
Days 42	7.45	7.98	7.81	7.62	0.10	0.764	0.105

MOS = mannooligosaccharide; SEM = standard error of means (each treatment included 48 chickens and consisted of 6 replicates); L = linear; Q = quadratic.

**Table 5 animals-09-00817-t005:** Intestinal immunoglobulins contents of Partridge Shank chickens fed diets supplemented with or without MOS (μg/mg protein).

Items	Control	0.5 g/kg MOS	1 g/kg MOS	1.5 g/kg MOS	SEM	*p*-Value	
						L	Q
Jejunum							
sIgA							
Days 21	7.37	7.62	8.43	8.31	0.24	0.110	0.708
Days 42	8.78	10.15	9.12	7.93	0.29	0.227	0.039
IgM							
Days 21	7.64	7.71	8.96	9.06	0.27	0.024	0.966
Days 42	11.41	12.14	11.06	9.38	0.42	0.053	0.140
IgG							
Days 21	108.17	138.09	143.99	168.33	7.46	0.004	0.823
Days 42	145.96	191.79	179.32	142.54	7.54	0.702	0.005
Ileum							
sIgA							
Days 21	8.25	9.35	10.41	8.34	0.33	0.914	0.022
Days 42	9.92	11.55	10.19	9.27	0.03	0.163	0.022
IgM							
Days 21	9.11	10.24	11.58	8.82	0.35	0.843	0.002
Days 42	10.91	14.87	12.90	10.81	0.49	0.505	0.001
IgG							
Days 21	133.93	145.24	175.10	122.59	7.14	0.940	0.018
Days 42	185.97	229.32	196.24	174.00	7.36	0.243	0.019

MOS = mannooligosaccharide; sIgA = secretory immunoglobulin A; IgM = immunoglobulin M; IgG = immunoglobulin G; SEM = standard error of means (each treatment included 48 chickens and consisted of 6 replicates); L = linear; Q = quadratic.

**Table 6 animals-09-00817-t006:** Intestinal antioxidant status of Partridge Shank chickens fed diets supplemented with or without MOS.

Items	Control	0.5 g/kg MOS	1 g/kg MOS	1.5 g/kg MOS	SEM	*p*-Value	
						L	Q
Jejunum							
T-SOD (U/mL)							
Days 21	1112	1179	1147	1195	26.42	0.345	0.175
Days 42	1180	1158	1155	1283	29.67	0.260	0.219
MDA (nmol/ mL)							
Days 21	6.99	3.76	3.53	3.92	0.44	0.004	0.024
Days 42	7.42	9.40	7.07	6.04	0.46	0.112	0.095
Ileum							
T-SOD (U/mL)							
Days 21	1040	1038	1185	1219	29.58	0.081	0.430
Days 42	1069	1071	1043	1162	19.89	0.698	0.196
MDA (nmol/ mL)							
Days 21	6.34	6.25	8.20	5.28	0.49	0.770	0.146
Days 42	7.10	13.56	10.50	6.51	0.73	0.244	<0.001

MOS = mannooligosaccharide; MDA = malondialdehyde; T-SOD = total superoxide dismutase; SEM = standard error of means (each treatment included 48 chickens and consisted of 6 replicates); L = linear; Q = quadratic.

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
