# Peer review of "Effects of Mannanoligosaccharide Supplementation on the Growth Performance, Immunity, and Oxidative Status of Partridge Shank Chickens"

_animals, 2019, doi:10.3390/ani9100817_

Round 1

Reviewer 1 Report

The MS analyses the effect of the supplementation with an enzimatic-obtained MOS in diet for broilers (0, 0,5, 1, 1,5 g/Kg) on some physiological and microbial parameters. The MS is well written, it offers a simple but effective background of the effect of synthesis-MOS on broilers health and investigate the effect of MOS obtained by from A konjac using β-mannanase. The results show that this MOS does not affect broilers health and is able to improve the oxidative status and obtain some positive tendency in the decrease of phatogens. At the same time only few differences were obtained regarding growth performances, while a significant increase in bursa weight was recorded with MOS inclusion. Since the intestine is a target organ for this kind of study, histological analysis and morphometry would have provided some more data and are surely necessary in further study (see Cutrignelli et al., 2017 DOI:10.1016/j.rvsc.2017.12.020 and Moniello et al., 2019 DOI:10.3390/ani9030086). In my opinion the present MS could be accepted after very minor revision.

Line 31-33 This sentence could be written in a more clear way since "immunoglobulin G (IgG, P<0.05)" seems to be a repetition

Line 66-67 This sentence could be written in a more clear way since "Konjact" seems to be a repetition

Line 89 Please explain how did you inactivated the enzyme 

Line 111 In my opinion it is better to write in an impersonal manner

Line 168 It --> it

Line 243 salmonella --> Salmonella 

Line 255-267 This paragraph could be improved for example explaining why the effects of MOS were mainly observed in jejunum IgG and ileum IgG, IgM and IgA

Reviewer 2 Report

There is nothing breath-taking about the results of this study. Basically confirms what is known of MOS responses in the literature. No real scientific contribution, but confirms that this particular MOS product works.

However, major changes are needed before it can be accepted.

Importantly, the appropriate statistical method for this design, where the responses are dose-dependent, is orthogonal polynomials to examine the linear and quadratic effects of graded levels of MOS. In this model, differences between individual treatments are NOT important. The authors are using superscripts to separate the significance between treatment means, which is not appropriate. In this statistical model, trends are relevant than treatment differences. Using linear or quadratic effects in responses would enable a much focussed presentation. The data should be presented accordingly in the Tables (i.e. L and Q probabilities) and, the 'ABSTRACT", 'RESULTS" and 'DISCUSSION' sections should be re-written. 

The paper is generally readable, but can benefit from some careful proof-reading. For example, L94 – three-layer cages?.

Some statements are confusing. See L110-112 – did they weight only 1 bird per cage;  L159 – only one bird as experimental unit? – if true, then these are fatal issues.

TABLE 2:

Data for 22-42 d redundant. Delete.

A body weight only 1.3 kg at 42d is very low. Why?.  Please comment. If breed effect, identify clearly – in the title etc.

In all Tables, indicate (as a foot note), no of birds per mean.

TABLE 6:

Follow decimal convention. e.g. 1112, not 1111.58 etc.

Round 2

Reviewer 2 Report

The major concerns have been addressed in the revised version

This manuscript is a resubmission of an earlier submission. The following is a list of the peer review reports and author responses from that submission.

Round 1

Reviewer 1 Report

The objective of the manuscript is to provide an evaluation in the use of plant-based MOS on intestinal immunity, antioxidant status, microbial population and performance parameters. It is highly recommended to the authors to review the use of English grammar. Unfortunately, this makes difficult to understand the ideas presented in the text. Additionally, it is important to verify the statistical analysis. In some results, there are statistical differences but no comparison between treatments. More detail comments are provided below with the intention of improving the quality of the manuscript.

Line – Comment

2 – Remove the article “the” from the title. Do you mean Mannanoligosaccharides?

2 – Do you mean “Mannooligosaccharides”? - plural? Check throughout the text.

17 – Check the sentence: “invading microorganism such as mannoligosaccharides”?  Mannan-oligosaccharides are microorganisms? – Check the structure of the sentence.

18 – Remove the article “the”

15 – 22 – Check grammar and sentence structure

27 – Do you mean 0, 0.5, 1 and 1.5 g of MOS per kg of diet? Please explain because the units “g/kg MOS” are confusing

28 – All the MOS concentrations improved the weight of the bursa?

29 – All the MOS concentration increased jejunal IgG, sIgA, and IgM?

30 – IgG was already mentioned.

31 – All the MOS concentration increased ileal T-SOD?

23 – 35 – The abstract requires major changes in the sequence of the results and proper description of the experimental treatments

40 – 44 – Check grammar and sentence structure

51 – “respectively”? – Remove the word if necessary

52 – “Fishes” – Please check grammar. Are you referring to different species of fish?

52 – Connect the sentence starting as “It has been demonstrated…” with the previous one.

58 – MOS are multiple – Therefore use plural.

61 – 74 – Check sentences structure, grammar, and connections between paragraphs. Unfortunately, it makes the text confusing for the reader.

79 – 83 – Check grammar. Include a reference for the enzymatic hydrolysis of KGM to obtain MOS. What was the source of KGM? Where did the authors buy it? This information is crucial because is describing the feed additive under evaluation.

85 – According to line 25 – It was 192 broiler chicks? Please verify

88 – Please verify the experimental treatments 0, 0.5, 1 and 1.5 g/kg MOS? Do the authors mean g/kg of diet?

89 – Basal diets? More than one

93 – Do you have a reference to support the use of 12 h of fasting? How does the author think this will affect the results of the experiment? Broilers are quite sensitive to feed removal, it is in fact considered a challenge situation to measure intestinal permeability and stress.

96 – FCR – Feed conversion ratio

106 – Days - plural

107 – Pen or cage?

108 – One bird from these candidate? – What do the authors mean? Please check all the sentence from grammar and structural mistakes.

108 – 116 – Check grammar and sentence structure. Do the authors mean 4450 x g?

112 – Bursa of Fabricius – Use the complete name.

119 – 121 – Use a connector to match with the previous paragraph. It is a different sample.

128 – Source of MacConkey and MRS agar?

134 – Are these the same samples described in line 119?

136 - Do the authors mean 4450 x g?

152 – Did the authors use each animal as an experimental unit for determination of mucosal immune and antioxidant parameters?

153 – Why there is no comparison between MOS treatments?

154 – Pen or cage? It cannot be both. Each cage was used as the experimental unit for performance parameters? Please be specific.

162 – It appears that MOS tend to reduce ADG during the 1-21d period – Negative results should also be included.

163 – The performance table should include results of the overall experimental period (1-42d). Also, avoid using abbreviations in the title of the tables.

163 – The ADG of all the treatments is low, based on the Arbor Acres performance objectives. Do the authors have an explanation for this?

168 – Bursa of Fabricius?  Use complete name throughout the text.

170 – There is a difference in the Bursa of Fabricius weight, but there is no indication of the difference between treatments.

173 – This information was not mentioned in the abstract. Please verify and include at least a sentence.

175 – Cecal

175 – 176 – Lactobacillus was or was not affected? Check and make the corresponding changes.

184 – In this section, the authors need to describe in detail the results presented in table 5. For example, it is not enough to comment that MOS increased ileal sIgA content in comparison to control without mentioning what concentration of MOS was better. Please check all the results and make the corresponding changes.

195 – In this section, the authors need to describe in detail the results presented in table 6. For example, it is not enough to comment that MOS decreased jejunal MDA without describing the comparison between treatments. Please check all the results and make the corresponding changes.

200 – Ileal MDA?

215 – 219 – Please explain this hypothesis in the text.

221 – 222 – Paraphrase

221 – 231 – Check grammar

233 – 235 – In this experiment, those results were not observed – Please check grammar and sentence structure.

236 – E. coli – Italicize

240 – Check this sentence. What do you mean under “health status”?

245 - Check this sentence. What do the authors mean with “dissimilar intestinal microorganisms?

268 – Check sentence – Grammar.

272 – “Fishes” – Please check grammar. Are you referring to different species of fish?

276 – 278 – Check sentence grammar and structure.

279 – 282 - Based on the data presented in the tables, the conclusion of the manuscript could be more specific. The authors had three different levels of MOS and the comparison between levels was not mentioned. In general, there is not conclusion regarding the level of MOS that could be more appropriate based on the actual data.

Reviewer 2 Report

The manuscript describes some phisiological effects of a partial inclusion of MOS in broilers diet. In the present study MOS was obtained by mean of enzimatic process by Aspergillus and was included at different concentrations (0,5, 1 and 1,5 g/Kg in the experimental diets). The idea of using MOS obtained in the described way sounds good in an evloronmentally friend approach, Nevertheless the methods and the results are too poor to give originality and a good scientific value to the paper.

Some comments:

-  in the session "2.3. Sample collection" (line 118-119) you mention that you sampled for histology.., but samples from which organs? and were are the results? In a study like this I am of the view that histological data at least on intestine are of essential importance!

- line 161-162  "MOS tended 161 to effect ADFI (average daily feed intake) during the whole experiment". This is not supported by data reported in tab.2 and, however, you shoul be more precise. Which groups?

- line 166-168 "the inclusion of MOS (Table 3) significantly increased 167 relative weight of bursa at 21 day (P = 0.019)" This is inconsistent with what is reported in the table and again you don't refer to groups but you talk about MOS in general 

- In the Tables it is not clear what the SEM and P values refers to. It is not indicated the significance for each value and Tukey’s multiple range tests results apart from that in table 5. Did you reported only the results of one way ANOVA?

-line 174 (and elsewhere) "affect" instead of "effect" 

- line 155 "cecal" instead of "cacal"

The discussion is too speculative since the method and the results are not sufficient to give the due indications on the use of this MOS.

Even if zootecnical and immune organs paramenters are indicative of nodifferences among experimental groups, you should improve the manuscript whit additional techniques such as histology, at least. 

Reviewer 3 Report

Comments to the Author

The aim of this study is to determine the effects of feeding Mannooligosaccharide to broiler chickens on growth performance, immunity and oxidative status. The subject is of interest and lies within the scope of the journal. However, it should be rejected because of the extremely low growth of the animals taking away the interest of the study. Experimental studies with broiler chicken must meet the commercial standards in performance otherwise they lose the practical interest of the research outcome. Also, a bad performance can interfere in the results obtained for immunity and oxidative status. Taking into account the data on performance in Table 2 of the manuscript broilers weight around 400 g at 21d of age (considering an initial bird weight of 40 g) while the normal weight for this age is around 900 g. For animals with 42 d of age a normal weight following the commercial tables is around 2800 g and your final weight is around 1280 g. Reaching less than the 50% of the potential growth of the animal the study must be rejected.

Reviewer 4 Report

The authors tackle and important topic – gut health of broiler chicks without the use of in feed growth promoting antimicrobials.  However, the manuscript is lacking in a number of ways.  First, the work is not original.  The potential use of hydrolysed konjac glucomannan as a prebiotic and for salmonella control has been investigated since the 2000’s (https://doi.org/10.1002/jfsa.2919 https://doi.org/10.4161/gmic.22728 ). 

Simple summary 

- line 18 can the authors clarify what feed additives they are referring to?

- line 18-19 can the authors clarify if they are including mannanoligosaccharides in that statement?

- line 20 can the authors provide evidence that the production of enzymes through fermentation ‘avoids’ pollution in the environment?  What makes enzyme production less polluting than chemical synthesis? Can the authors please reference the relevant literature to support such a statement. 

The introduction is very confusing for the reader.  The authors use the term MOS quite loosely, they need to be more specific which structural forms and sources (yeast / fungal versus plant) of MOS have which properties.  The work is not sufficiently referenced for the reader to get an accurate understanding of which form of MOS has which mode of action.  The authors need to focus on plant MOS and specifically the hydrolysed konjac form and the associated literature.  The yeast literature should not feature prominently in the introduction as its misleading for the aforementioned reasons. 

Do the authors consider their specific mannanoligosaccharide(s) to be unique in chemical structure or method of manufacturing?   If that is the case, then the method of manufacturing and specific structural characterisation has to be much further defined for the reader to understand the difference between this and other glycosylated glucomannan structures.  At a minimum I would expect the authors to detail the purity and complexity of the mannanoligosaccharide by size exclusion chromatography and H-NMR and glycosyl composition by GC-MS.  The reader would need to know if the product is indeed composed of oligomers or polymers.   

Did the authors screen the properties of the hydrolysed konjac MOS in vitro for immune activity, salmonella binding or other?  What is the activity measurement of the product? 

Does the final mannanoligosaccharide product retain it’s enzymatic activity?  Have the authors considered that the B-mannanase enzyme may be releasing energy from the cereal feed ingredients in the diets?

Line 87 the use of 6 replicates seems low, especially with no replication of the experiment.  Can the authors provide the Power number for the experimental design rationale? 

Lines 116-119 Sample collection for histopathology

The authors need to remove as they do not present any data relating to these samples. 

Microflora Population Measurement

Semi-selective media were employed for microbial enumeration.  What confirmatory tests, if any, were performed to confirm that the colonies on the MacConkey plates were E. coli and not just coliforms; same for Salmonella on Bismuth sulphite agar and MRS agar for Lactobacillus?   There seems to be a large presumption by the authors on the specificity of the microbiological media.  Why were these bacteria considered and not others? Did the authors have a mode of action in mind?

Determination of the mucosal immune and antioxidant parameters

The authors should justify the selection of the parameters tested.  Did the authors consider looking at pro- and anti-inflammatory signalling markers? 

Results

What was the mortality rate by treatment in the study? 

Tables 2, 3, 4 and 6 do not display the Tukeys means comparisons.  It makes it difficult for the reader to compare treatments for statistical difference or ‘trend’ without this information. 

Table 2.  There are no statistical differences between treatments

Table 3. There is a statistical difference in the bursa weight at 21 days

Table 4.  There are no statistical differences between treatments.  The salmonella results are very high for a research facility.  Do the authors typically see such a high salmonella cecal count in a controlled condition.  Did the authors test the feed?  No information is presented on the sanitisation of the facility before the study. Could this be provided?

Table 5. There are statistical differences between treatments for Jejunum IgG, Ileal sIgA, IgG and IgM.

Table 6.  There are statistical differences between treatments but no comparison test results in superscript so difficult to interpret.  Did the authors consider using an alternative post-hoc test?  Perhaps Dunnetts to compare treatments versus the Control. 

Discussion

It is difficult to interpret the results of this study.  Why would the bursa weight be increased in the MOS treatments when there is a high colonisation of E. coli and salmonella present in all treatments including the Control.  Why is the bursa weight highest with the lowest level of MOS inclusion?  Did the authors perform a regression analysis vs the other parameters measured to see if this bursa weight increases can be explained by other immune parameter shifts? 

Line 233 –235 do the authors have evidence that their MOS ‘promotes growth of Bifidobacteria, ….improves mucosal integrity? 

Line 235-237 It is well documented that MOS from yeast cell wall can block type 1 fimbriae of G-ve bacteria.  However, it is more controversial for plant derived MOS.  The oligosaccharide structure determines binding capability https://doi.org/10.4161/gmic.22728

Conclusions

The authors need to be more specific with their findings.  First, it would help to add the means comparisons from the post-hoc test to the data tables to allow the reader interpret the results.